# Natural Antibacterial Compounds with Potential for Incorporation into Dental Adhesives: A Systematic Review

**DOI:** 10.3390/polym16223217

**Published:** 2024-11-20

**Authors:** Ana Catarina Sousa, Paulo Mascarenhas, Mário Polido, Joana Vasconcelos e Cruz

**Affiliations:** 1Instituto Universitário Egas Moniz (IUEM), Egas Moniz School of Health & Science, 2829-511 Caparica, Portugalpmascarenhas@egasmoniz.edu.pt (P.M.); mpolido@egasmoniz.edu.pt (M.P.); 2Egas Moniz Center for Interdisciplinary Research (CiiEM), Egas Moniz School of Health & Science, 2829-511 Caparica, Portugal

**Keywords:** natural products, antibacterial effect, antibacterial agent, dental adhesive, biocompatibility

## Abstract

Dental adhesives are essential in modern restorative dentistry and are constantly evolving. However, challenges like secondary caries from bacterial infiltration at the adhesive–tooth interface persist. While synthetic antibacterial agents in adhesives show promise, safety concerns have shifted interest toward natural options that are biocompatible, sustainable, and effective. Therefore, this study evaluated whether natural antibacterial compounds in dental adhesives can provide effective antimicrobial activity without compromising their integrity. This systematic review followed PRISMA 2020 statement guidelines. Four databases were screened, PubMed, Scopus, EMBASE, and Web of Science, without language or publication date restrictions until July 2024. The selection criteria were in vitro studies in which natural antimicrobial substances were incorporated into dental adhesives and the resulting composites were tested for their antibacterial and physicochemical properties. A quality assessment was conducted on the selected studies. Most of the studies reviewed reported significant antibacterial activity while retaining the adhesive’s integrity, generally achieved with lower concentrations of the natural agents. Higher concentrations increase the antimicrobial effectiveness but negatively impact the adhesive’s properties. This review highlights the promising role of natural antibacterial compounds in enhancing the functionality of dental adhesives while also pointing to the need for continued research to address current challenges.

## 1. Introduction

Dental adhesive technology has revolutionized many aspects of restorative dentistry, allowing for a more conservative approach as there is no need to remove healthy tooth structures to provide mechanical retention [1]. In recent decades, dental adhesive systems have continuously evolved to improve bonding effectiveness with enamel and dentin, aiming to create durable restorations with well-sealed margins [2]. A major breakthrough in achieving direct adhesion to dental tissues came in 1955 when Buonocore demonstrated that acid etching could significantly improve the bonding of restorations to enamel [3]. Dentin bonding is more complex and relies on the formation of the “hybrid layer”, first reported by Nakabayashi in 1982 [4]. His research also underscored the importance of using monomers that contain both hydrophobic and hydrophilic groups, promoting better infiltration into the dentinal tubules and enhancing the adhesive bond [4]. Currently, two different strategies can be employed in resin bonding procedures: the etch-and-rinse technique (E&R) and the self-etch (SE) technique [5].

Nonetheless, despite the significant advancements in dental adhesives, several obstacles remain to be overcome, as the failure and subsequent need for the replacement of resin composite restorations are still major concerns [6]. Secondary caries is described in the literature as the most common reason for replacing existing restorations and usually occurs due to the penetration and subsequent proliferation of cariogenic bacteria along microgaps between tooth tissue and the restoration [7,8,9]. The contraction stress resulting from the polymerization shrinkage of dental adhesives has been demonstrated to contribute to the formation of marginal microgaps at the adhesive–tooth interface. This phenomenon permits the progressive infiltration of salivary fluids and microorganisms into the restoration [10,11]. Furthermore, the hydrolysis of the adhesive interface compromises the marginal integrity of composite restorations, thereby facilitating bacterial microleakage [12]. The biofilm accumulation on composite restorations represents a further issue, as these materials tend to accumulate more biofilm than other restorative materials [13]. Moreover, the use of minimally invasive techniques in dentistry has increased. In some cases, carious tissue removal is not complete, particularly in deep lesions, to prevent damage to the pulp and preserve the tooth structure [14]. However, this methodology may result in the retention of active bacteria within the dentin, which could ultimately lead to the failure of the restoration [15].

In light of this problem, the research and development of antibacterial dental materials that inhibit the growth and accumulation of oral bacteria represent a promising avenue for reducing the incidence of secondary caries [16,17]. Dental adhesive systems are in direct contact with the tooth surface, making them an ideal material to possess antibacterial activity, as this area is susceptible to bacterial contamination [18]. Various studies have incorporated antimicrobial agents in dental adhesives, of which quaternary ammonium monomers [19,20,21], silver nanoparticles [17,22], zinc methacrylates [23,24], and chlorhexidine [25] are some of the most commonly described. The first successful antibacterial dentin primer resulted from the incorporation of MDPB. This antibacterial quaternary ammonium monomer is currently present in the composition of a commercial dental adhesive, ClearfilTM SE Protect (Kuraray Noritake Dental Inc., Tokyo, Japan) [20]. However, there has been a growing concern regarding the biocompatibility and cytotoxic effects of some synthetic bactericidal agents used in dental materials [26].

As a result, the interest in applying natural antibacterial products in dental materials has increased, as they constitute a promising alternative to synthetic antibacterial agents due to their therapeutic properties [27]. Natural compounds have been demonstrated to be effective in combating microbial activity. Furthermore, they are noted for their enhanced biocompatibility, which ensures better integration with biological tissues and minimizes the risk of adverse effects when administered at appropriate doses [28]. Additionally, these compounds constitute a more sustainable and cost-effective option, which aligns with a global shift towards more eco-conscious healthcare solutions [29]. A natural product is defined by Umaru (2023) as a “substance synthesized by plants or animals, or chemical substances found in nature with particular pharmacological effects” [30].

This study aimed to conduct a systematic review of in vitro studies that evaluated the antibacterial efficacy of dental adhesive systems incorporating natural antibacterial compounds. The hypothesis to be tested was whether natural antibacterial compounds, when integrated into experimental or commercial dental adhesive systems, can provide an effective antimicrobial activity without compromising the physicochemical properties of the adhesive.

## 2. Materials and Methods

The present systematic review was conducted in accordance with the guidelines of the PRISMA 2020 statement [31]. The protocol for this study was submitted to PROSPERO International Prospective Register of Systematic Reviews and registered under nr. CRD42024571079.

### 2.1. Research Question and PICO Method

The research question was defined using the PICO framework:

P (Population): Experimental and commercial dental adhesive systems.

I (Intervention): Incorporation of natural antimicrobial substances into dental adhesive systems.

C (Comparison): Control groups of dental adhesive systems without incorporating natural antimicrobial substances.

O (Outcome): Antibacterial efficacy and physicochemical properties (specifically, degree of conversion and bond strength) of modified dental adhesives evaluated in vitro.

As a result, the following research question was formulated: What natural substances can be added to dental adhesives to improve their antibacterial properties without compromising their integrity and performance?

### 2.2. Search Strategy

The literature search was carried out until 18 July 2024. The electronic bibliographic databases selected included PubMed, Scopus, EMBASE, and Web of Science. There were no language or publication date restrictions, and an individualized search strategy was developed for each database (Table 1). All references resulting from the initial search were imported into the online platform *Rayyan—Intelligent Systematic Review* (Rayyan Systems Inc., Cambridge, MA, USA) [32] to remove duplicates.

### 2.3. Inclusion Criteria

The following inclusion criteria were applied: controlled in vitro studies incorporating natural antimicrobial substances into dental adhesive systems and testing the modified adhesives’ antibacterial efficacy and physicochemical properties, specifically the degree of conversion and bond strength evaluations. The studies were required to include control groups.

### 2.4. Screening and Data Extraction

Two reviewers, working independently, undertook a comprehensive analysis and selection process for all titles and abstracts of the articles retrieved from the search strategy. This was conducted following the pre-established inclusion criteria. Those articles that appeared to meet the eligibility criteria were selected for a full-text analysis. This process was conducted independently and in duplicate. In the event of any discrepancies regarding the eligibility of the studies included, a consensus was reached through discussion, and when necessary, a third reviewer from the team was consulted.

The data extracted from each selected study included the following: authors and publication date; natural antibacterial compounds used, as well as the concentrations applied and the materials they were incorporated into; methodology of the antibacterial activity assessment; specimens used; microorganisms tested; physicochemical properties of the modified adhesives tested; and main conclusions.

### 2.5. Risk of Bias Assessment

Two independent reviewers conducted the quality assessment using the RoB-DEMAT tool [33], a recently developed risk-of-bias tool designed for pre-clinical dental materials research. The tool assesses four domains: bias in planning and allocation, bias in sample/specimen preparation, bias in outcome assessment, and bias in data treatment and outcome reporting. In the event of a discrepancy between the two reviewers’ assessments, a third team member was consulted to facilitate a discussion and reach a consensus. The risk of bias (RoB) analysis of each study is presented in Appendix A, with each signaling question classified as “reported”, “insufficiently reported”, “not reported”, or “not applicable”, in alignment with the RoBDEMAT tool guidance. The data from Appendix A were adapted according to the RoBDEMAT tool classification guidance, with “reported” classified as low risk, “insufficiently reported” as medium risk, and “not reported” as high risk. These adapted data were then uploaded to the Robvis online tool [34], which generated a Traffick Light Plot and a Summary Plot of the studies.

## 3. Results

### 3.1. Study Selection

Figure 1 visually represents the study selection process, presented as a flowchart. A total of 15,255 potentially relevant records were identified across the four databases. After removing the duplicates, 11,396 studies remained for the title and abstract screening. Out of these, 25 records were selected for full-text reading. This resulted in the exclusion of twelve studies since two did not incorporate the natural compound into a dental adhesive system, four did not assess any physicochemical properties of the adhesive, and six did not perform degree of conversion measurements. Therefore, a total of 13 studies were included in this systematic review.

### 3.2. Articles’ General Features

Table 2 summarizes the general characteristics of the studies considered in this review in chronological order. The publication years of the articles varied from 2012 to 2022.

The studies examined a diverse range of natural antibacterial agents, comprising 13, most of which were of the phenolic compound type. These included epigallocatechin-3-gallate (EGCG), found in green tea [35]; thymol, a key component of thyme essential oil [36]; and proanthocyanidin [37,38], quercetin [39], and apigenin [38], which are commonly present in various plant-based sources. Cashew nut shell liquid (CNSL) and its constituents, anacardic acid and cardol, were also part of this group [40]. The other natural agents studied were nisin, an antimicrobial peptide produced by Lactococcus lactis [41,42]; an essential oil derived from the Butia capitata palm tree [43]; chitosan, a natural polysaccharide biopolymer derived from chitin [44,45]; arginine, an amino acid naturally produced by the human body [46]; and tt-farnesol, a natural sesquiterpene alcohol found in propolis [18,38].

The antibacterial agents were incorporated into both commercial and experimental dental adhesive systems. Common commercial adhesives used included Adper™ Single Bond 2 [35,39,41,44], Adper™ Scotchbond™ Universal [18], Single Bond Universal [42], and Ambar APS [40]. Some studies utilized experimental formulations to adjust the adhesive properties by adding the natural compound directly to the adhesive [36,37,43,46] or incorporating it into an experimental primer [38,45].

The concentration of the antibacterial agents exhibited considerable variation across studies, reflecting the differing antimicrobial strengths observed between compounds. Most studies also tested different concentrations of the same agents to achieve an optimal balance between the highest antimicrobial efficacy and maintaining the adhesive’s properties.

**Table 2 polymers-16-03217-t002:** Studies’ characteristics.

Author	Natural Antibacterial Agent	Incorporation Material	Concentrations	Antibacterial Effect Assessment	Specimens	Microorganisms Tested	Adhesive Properties Tested	Main Conclusions
Du et al. (2012) [35]	Epigallocatechin-3-gallate (EGCG)	Adper^TM^ Single Bond 2 (SB.) (3M ESPE, St. Paul, MN, USA)	100, 200, and 300 μg/mL	Direct contact test and SEM evaluation	Adhesive + composite discs	*Streptococcus* *mutans*	Microtensile bond strength and degree of conversion	Incorporation of EGCG at the concentration of 200 μg /mL exhibits antibacterial activity whilst maintaining the adhesive integrity.
Elsaka (2012) [44]	Chitosan	Adper^TM^ Single Bond 2 (3M ESPE, St. Paul, MN, USA)	0.12%, 0.25%, 0.5%, and 1% (*w*/*w*)	Direct contact test	Adhesive discs	*Streptococcus* *mutans*	Microtensile bond strength, degree of conversion, and viscosity	Experimental adhesive resin containing chitosan shows antibacterial effect. Incorporating 0.12% (*w*/*w*) chitosan does not adversely affect the adhesive properties.
Peralta et al. (2013) [43]	*Butia Capitata* essential oil	Experimental adhesive	1 mol%	Bacterial viability (colony counts) and pH supernatant	Enamel discs + primer and adhesive	Human saliva (*Streptococcus* *mutans*, lactobacilli, aciduric bacteria, and total microorganisms)	Microtensile bond strength, degree of conversion, water sorption, and solubility	*B. Capitata* oil presents similar antibacterial activity to a commercial antimicrobial adhesive, but also to the control. Adhesive properties were retained; however, there was a decrease in bond strength after 6 months.
Geraldeli et al. (2017) [46]	Arginine	Experimental adhesive	7%	Direct contact test and CLSM	Adhesive discs	*Streptococcus* *mutans* and *Streptococcus* *gordonii*	Microtensile bond strength, degree of conversion, ultimate tensile strength, flexural strength, flexural modulus, and Knoop hardness	Adhesive system containing 7% arginine exhibits antibacterial effects, while retaining physical and mechanical properties.
Yang et al. (2017) [39]	Quercetin	Adper^TM^ Single Bond 2 (SB.) (3M ESPE, St. Paul, MN, USA)	100, 500, and 1000 μg/mL	CLSM and XTT assay	Adhesive + composite discs	*Streptococcus* *mutans*	Microtensile bond strength, degree of conversion, and nanoleakage	Adhesive modification with 500 μg/mL quercetin showed a balanced status of the antibacterial ability and adhesive properties.
Su et al. (2018) [41]	Nisin	Adper^TM^ Single Bond 2 (3M, St. Paul, MN, USA)	1%, 3%, and 5% (*w*/*v*)	Film contact test, agar diffusion test, XTT assay, and CLSM	Adhesive + composite discs Paper discs impregnated with the adhesive (ADT)	*Streptococcus* *mutans*	Microtensile bond strength and degree of conversion	The nisin-incorporated adhesive significantly inhibits the growth of *Streptococcus mutans* and its biofilm. However, concentrations above 1% exhibit a decrease in bond strength.
Rezaeian et al. (2019) [36]	Thymol	Experimental adhesive	5 wt%	Direct contact test (based on ASTM E 2180–07 [47])	Adhesive discs	*Streptococcus* *mutans*	Microshear bond strength, degree of conversion, and flexural and viscoelastic properties	The thymol-incorporated adhesive showed appropriate antibacterial activity and comparable physico-mechanical properties to the control adhesive.
Dias et al. (2020) [37]	Proanthocyanidin	Experimental adhesive	1 wt%, 2 wt%, 4.5 wt%, and 6 wt%	Bacterial growth and MTT assay	Adhesive discs	*Streptococcus* *mutans*	Microtensile bond strength, degree of conversion, water sorption, and solubility	The incorporation of proanthocyanidin did not promote an antibacterial effect in the adhesive.
Leyva del Rio et al. (2020) [18]	Tt-farnesol	Adper^TM^ Scotchbond Universal (3M ESPE, St. Paul, MN, United States)	0.38%, 1.90% and 3.80% (*v*/*v*)	Colony-forming units, biofilm dry weight, production of extracellular insoluble polysaccharides, and SEM	Adhesive + composite discs	*Streptococcus* *mutans*	Microtensile bond strength, degree of conversion, and hybrid layer permeability	Tt-farnesol increased the antibacterial activity of the universal adhesive system. However, the degree of conversion and bonding effectiveness of the adhesive were altered.
Zhao et al. (2020) [42]	Nisin	Single Bond Universal (3M, St. Paul, MN, U.S.A.)	1%, 2%, and 3% (*w*/*v*)	CLSM, qRT-PCR, PSA, and LDH	Adhesive + composite discs	*Streptococcus* *mutans* and saliva-derived multispecies	Microtensile bond strength and degree of conversion	The incorporation of 3% (*w*/*v*) nisin in the adhesive achieved a substantial antibacterial activity without compromising the bonding properties.
Ribeiro et al. (2021) [38]	Apigenin Proanthocyanidin Tt-farnesol	Experimental primer	4.5% proanthocyanidin, 1 mM apigenin, and 1 mM apigenin + 5 mM tt-farnesol	Hardness loss of enamel and dentin at the restorative margin	Enamel and dentin restorations	*Streptococcus* *mutans*	Microtensile bond strength, degree of conversion, nanoleakage, water sorption, and solubility	Integration of apigenin and proanthocyanidin in a dental adhesive system showed promising results in preventing secondary caries in enamel and dentin, without compromising the adhesive physical properties. The association of apigenin + tt-farnesol decreased bond strength after 1 year and was not effective in reducing hardness loss in enamel.
de Oliveira Souza et al. (2022) [40]	Cashew nut shell liquid (CNSL) Anacardic acid Cardol	Ambar APS (FGM, Joinville, SC, Brazil)	15 μg/mL (each compound separately)	Direct contact test	Adhesive discs	*Streptococcus* *mutans* and *Candida albicans*	Microtensile bond strength, degree of conversion, elastic modulus, flexural resistance, water sorption, and solubility	All 3 compounds showed antibacterial activity without jeopardizing the adhesive performance.
Yao et al. (2022) [45]	Carboxymethyl chitosan (CMC)	Experimental primer	5, 10, and 20 mg/mL	Direct contact test, XTT assay and CLSM	Primer and adhesive + composite discs	*Streptococcus* *mutans*	Microtensile bond strength and degree of conversion	The incorporation of 20 mg/mL CMC obtained the highest antibacterial activity and did not adversely affect the adhesive properties.

ADT = Agar diffusion test; ASTM E 2180–07 = Standard Test Method for Determining the Activity of Incorporated Antimicrobial Agent(s) in Polymeric or Hydrophobic Materials; EGCG = Epigallocate-chin-3-gallate; CLSM = Confocal laser scanning microscopy; CMC = Carboxymethyl chitosan; CNSL = Cashew nut shell liquid; LDH = Lactate dehydrogenase enzymatic method; MTT = 3-(4, 5-dimethylthiazolyl-2)-2, 5-diphenyltetrazolium bromide; PSA = Phenol–sulfuric acid method; qRT-PCR = Reverse transcription-quantitative polymerase chain reaction; SEM = Scanning electron microscopy; XTT = 2,3-bis-(2-methoxy-4-nitro-5-sulphenyl)-(2H)-tetrazolium-5-carboxanilide.

A variety of methods were employed to evaluate the antibacterial activity of the modified adhesives, with direct contact tests being the most frequently used among the studies, followed by confocal laser scanning microscopy (CLSM) and XTT (2,3-bis-(2-methoxy-4-nitro-5-sulphenyl)-(2H)-tetrazolium-5-carboxanilide) assays. Scanning electron microscopy evaluation (SEM), the measurement of extracellular polysaccharide production, and hardness loss of enamel and dentin are other tests employed to determine the adhesives’ ability to inhibit bacterial growth and biofilm formation.

For these assays, specimens were chosen to simulate clinical conditions. Most studies conducted antibacterial tests on adhesive discs or a combination of adhesive and composite discs. Only two studies used tooth tissue as substrates, specifically enamel discs coated with adhesive [43] and enamel and dentin restorations [38]. The latter specimens provide a more realistic assessment of how the adhesive system would perform in a dental setting.

*Streptococcus mutans* was the primary microorganism tested in all studies, reflecting its role in developing dental caries. Additionally, some studies evaluated the modified adhesives’ antibacterial effects against other microorganisms, including lactobacilli and aciduric bacteria present in human saliva [43], *Streptococcus gordonii* [46], *Candida albicans* [40], and the total microbial population in human saliva [42,43], to assess the broader antimicrobial spectrum of the agents.

The studies evaluated several key adhesive properties to ensure the adhesive systems maintain functionality after incorporating natural antibacterial agents. All studies performed a degree of conversion and bond strength evaluations, as these were part of the eligibility criteria. Furthermore, some studies assessed additional properties like flexural strength, modulus of elasticity, nanoleakage, viscosity, water sorption, and solubility.

### 3.3. Studies’ Patterns and Important Outcomes

#### 3.3.1. Antibacterial Effectiveness

In general, the majority of the natural compounds studied showed significant antibacterial activity when incorporated into dental adhesives, which varied depending on the concentration and agent used. Studies that tested different concentrations of these natural agents revealed a pattern of higher antimicrobial effectiveness at increased doses [18,35,39,41,42,45], with some compounds only showing significant antibacterial activity above certain concentrations. This was the case of EGCG [35], nisin [42], and tt-farnesol [18], which only exhibited significant activity above 200 μg/mL, 2% (*w*/*v*) for *Streptococcus mutans*, and 3% (*w*/*v*) for saliva-derived multispecies, and 1.90% (*v*/*v*), respectively. The only exception to this pattern was the study by Elsaka (2012), which found that chitosan provided a statistically similar antibacterial effectiveness across all tested concentrations, showing no noticeable increase in the antibacterial efficacy with higher doses [44].

Nevertheless, two studies did not demonstrate a significant antibacterial effect, as the results were not different from those of the control group. This was observed with the incorporation of proanthocyanidin by Dias et al. (2020) [37] and *Butia capitata* oil [43]. Additionally, the combination of apigenin + tt-farnesol reduced hardness loss in dentin but not in enamel [38].

#### 3.3.2. Adhesive Integrity

The reviewed studies showed that natural antibacterial compounds can be successfully incorporated into dental adhesives without compromising their crucial properties. In cases where different concentrations of the natural agents were tested, lower concentrations tended to preserve the adhesive’s properties, while higher doses often compromised adhesive integrity [39,41,42,44,46]. However, some exceptions deviated from this pattern, as EGCG was able to retain bonding properties even at higher concentrations and increase microtensile bond strength at lower doses [35]. Both carboxymethyl chitosan (CMC) [45] and proanthocyanidin [37] did not report differences in adhesive performance at higher concentrations.

On the contrary, the incorporation of tt-farnesol [18] significantly reduced the degree of conversion and bond strength, regardless of its concentration. The microtensile bond strength was also reduced with the association of apigenin + tt-farnesol after 1 year of aging [38], and with *Butia capitata* oil after 6 months [43]. However, *Butia capitata* oil presented bond strength values after aging that were statistically similar to those of other commercial adhesives [43].

#### 3.3.3. Long-Term Stability

While the studies showed promising short-term results, the long-term performance of these modified adhesives needs further investigation, as only five studies evaluated the antibacterial activity of the modified adhesives after aging, and only six studies assessed their properties over time.

Regarding antimicrobial effectiveness, the aging period of the studies ranged from 5 days to 1 year. Despite some reduction over time, the adhesives generally retained their antibacterial activity at higher concentrations of the natural agents [35,36,39,44]. Tt-farnesol, however, reported increased antimicrobial effectiveness after 5 days [18].

After ageing, long-term adhesive stability was evaluated through microtensile bond strength and nanoleakage evaluations. The results were varied: EGCG [35] and CMC [45] maintained bond strength across all tested concentrations, while only 500 μg/mL quercetin [39] and proanthocyanidin above 1 wt% [37] showed similar outcomes. In contrast, *Butia capitata* oil [43] and the combination of apigenin + tt-farnesol [38] exhibited reduced bond strength after 6 months and 1 year, respectively. Nanoleakage assessments revealed no increase after 1 month for any concentration of quercetin [39], but there was a significant increase with apigenin and the apigenin + tt-farnesol combination [38].

### 3.4. RoB Analysis of the Studies

The results of the risk of bias analysis and the overall score for each selected study and each domain are presented in Figure 2 and Figure 3, respectively. All studies exhibited a considerable risk of bias, as none included a sample size calculation or blinding of the test operator. However, all studies provided an adequate control group and statistical analysis, with only one study [18] exhibiting insufficient reporting of identical experimental conditions across groups. The degree of bias related to the standardization of samples, testing procedures, and outcome reporting exhibited variability among the studies, ranging from low to medium risk. The application of sample randomization was limited to the two studies [38,43] that utilized teeth as specimens to evaluate antibacterial activity.

## 4. Discussion

### 4.1. Antibacterial Effectiveness

Dental caries is the most prevalent disease worldwide, estimated to affect over 2.5 billion people [48]. Resin composite restorations are a widely chosen treatment to replace decayed tissue, given that they offer excellent aesthetics and a less invasive cavity preparation [49]. Nonetheless, composite restorations often have a high failure and replacement rate, mostly attributed to bacterial microleakage at the tooth–restoration margins, leading to secondary caries [50]. Various synthetic antimicrobial agents have been studied and incorporated into experimental and commercial dental adhesives to overcome this problem, such as quaternary ammonium compounds, metallic nanoparticles, and chlorhexidine [16].

However, despite being a less explored source of antibacterial agents for incorporation into dental materials, natural products could provide an alternative of interest to synthetic materials due to their biocompatibility, sustainability, and therapeutical properties [51]. Furthermore, the growing awareness of the impact of consumer choices on health and the environment has prompted a notable shift in consumer behavior, with an increasing preference for natural and biological foods and materials. This indicates a heightened consciousness among consumers regarding the implications of their choices on their well-being and the planet [29]. This systematic review found consistent evidence across most of the studies selected that incorporating natural antibacterial compounds into dental adhesives can effectively inhibit bacterial growth and biofilm formation, especially from *Streptococcus mutans*, since it was the most tested microorganism. This suggests that these natural agents hold promise for enhancing the antimicrobial properties of dental adhesives, potentially reducing the occurrence of secondary caries and increasing the lifetime of composite restorations [52].

This review evaluated 13 different natural substances across 13 studies. Focusing on their antibacterial mechanisms of action, most of the compounds primarily exert their effects through membrane disruption, which causes structural damage that compromises membrane integrity and increases permeability, ultimately leading to cell lysis [53,54,55,56,57,58,59,60,61]. Compounds such as EGCG, quercetin, proanthocyanidins, and apigenin also inhibit essential enzymatic activities, disrupting bacterial energy metabolism and impairing the cell’s ability to maintain vital functions [53,55,57,59]. Arginine modulates pH levels, inhibiting the growth of acid-producing bacteria [62]. Finally, nisin, a bacteriocin, inhibits bacterial growth by creating pores in the cell membrane and disrupting cell wall biosynthesis through a targeted interaction with lipid II [63]. Membrane disruption and enzyme inhibition represent two of the most effective mechanisms of antibacterial action since they exhibit broad-spectrum efficacy, targeting both Gram + and Gram − bacteria [64].

Two reviewed studies failed to demonstrate a significant antibacterial activity of the dental adhesive modified by incorporating a natural agent.

*Butia capitata* oil demonstrated similar antibacterial activity to its control despite presenting similar bacterial growth inhibition values to a commercial antibacterial adhesive containing MDBP, a quaternary ammonium monomer [43]. In this study, Peralta et al. (2013) only tested a single essential oil concentration, which may lead to the hypothesis that a higher concentration could possess stronger antimicrobial effects. This is consistent with the findings of the other studies reviewed, which also indicated that higher concentrations of essential oils may have stronger antimicrobial effects. In contrast to those results, Peralta et al. (2017) conducted another study involving the incorporation of *Butia capitata* oil in a dental adhesive system, using the same concentration tested in 2013 (1 mol% of the essential oil) and the same experimental adhesive formulation [61]. However, this time, the adhesive incorporated with the natural agent demonstrated a significantly higher antimicrobial performance than the experimental control adhesive and, once again, statistically similar values compared to the same commercial antibacterial adhesive used in 2013 [61]. It is worth mentioning that the study from Peralta et al. (2017) was excluded from this review for not assessing the modified adhesive physicochemical properties, which was part of the inclusion criteria.

The incorporation of proanthocyanidin in an experimental dental adhesive by Dias et al. (2020) was also unable to enhance the antibacterial activity of the adhesive since all concentrations tested (1 wt%, 2 wt%, 4.5 wt%, and 6 wt%) presented similar levels of cell growth and metabolic activity inhibition of biofilm-forming bacteria between them and the control [37]. These results are contradicted by Ribeiro et al. (2021), who showed that the incorporation of 4,5% proanthocyanidin in an experimental primer was able to reduce hardness loss in dentin and enamel when compared to the control, thus showing an increase in the antibacterial effect [38]. Nevertheless, the findings of Dias et al. (2020) also showed that, despite having similar values, all groups possessed high antimicrobial effectiveness, even the control group, as a comparison was established with a commercial dental adhesive. Hence, it could be hypothesized that the antibacterial effect presented in the control was due to the experimental adhesive’s low pH, ranging from 0.6 to 0.8, whereas the commercial adhesive also tested had a pH of 5.4 [65]. However, the antimicrobial effect achieved by the low pH of dental materials is considered limited in terms of bactericidal spectrum and durability [66].

Furthermore, Ribeiro et al. (2021) demonstrated that the combination of apigenin and tt-farnesol exhibited antibacterial efficacy in dentin, as evidenced by reduced hardness loss. However, this combination did not demonstrate the same effect in enamel. Nevertheless, the same study demonstrated that incorporating apigenin (at the same concentration) into an experimental primer reduced hardness loss in dentin and enamel [38]. Another study in this review showed that tt-farnesol could reduce *Streptococcus mutans* viability when incorporated into a dental adhesive at a concentration above 1.90% (*v*/*v*). However, since the specimens used were composed of only adhesive + composite discs, it was impossible to understand the effects of tt-farnesol on enamel and dentin [18].

Chitosan was evaluated in two different studies included in this review. Elsaka (2012) incorporated a chitosan solution (chitosan powder dissolved in acetic acid) into a commercial dental adhesive, while Yao et al. (2022) integrated carboxymethyl chitosan (CMC), a derivate of chitosan that is modified through carboxymethylation, into an experimental primer. Both studies found that modifying a dental adhesive system with chitosan can provide an effective antibacterial effect against *Streptococcus mutans* [40,44]. Likewise, nisin was included in two of the studies reviewed, both of which reported an increase in antimicrobial activity due to incorporating the peptide into a dental adhesive [41,42].

Most studies have evaluated the antibacterial activity of modified dental adhesives solely against *Streptococcus mutans*. However, only four studies have included additional microorganisms in their analysis [40,42,43,46]. *Streptococcus mutans* is widely acknowledged in the literature as a principal cariogenic agent, significantly contributing to the development of dental caries. Consequently, the inhibition of this microorganism is of paramount importance for the efficacy of an antimicrobial dental material [67]. Nonetheless, it is also important to evaluate this effectiveness against other microorganisms involved in the formation of dental caries, such as *Streptococcus sorbinus*, *Lactobacillus* spp., and *Actinomyces* spp, particularly in biofilms, since they exhibit a greater resistance to antibiotics compared to planktonic cells [68,69].

Despite the promising results regarding the antibacterial activity of these natural compounds, it is important to mention that only two studies used tooth substrates to perform these assays. Peralta et al. (2013) used bovine enamel, and Ribeiro et al. (2021) used human dentin and enamel. Without tooth substrates, it is impossible to accurately evaluate the effectiveness of antibacterial agents in preventing the formation of secondary caries [70]. Therefore, further investigations are necessary to assess these natural compounds’ potential antibacterial effectiveness properly. Moreover, using different methodologies to test the antibacterial activity makes comparing the results across different studies challenging, highlighting the need for a standardized method [71].

### 4.2. Adhesive Integrity

One of the critical challenges in incorporating antibacterial agents into dental adhesives is preserving their integrity and stability, as any deterioration of the material’s physicochemical properties could compromise the adhesive interface and lead to restoration failure [72]. Therefore, assessing these properties is crucial in developing dental materials [73], which is why it was included in the eligibility criteria of this review, specifically degree of conversion and bond strength evaluations. A high degree of conversion has been correlated to a greater bond strength, biocompatibility, and restoration durability since an incomplete polymerization of the adhesive can result in unreacted monomers within the hybrid layer, creating a porous structure with reduced sealing ability and increased permeability, making it more susceptible to degradation [74,75]. A bond strength test evaluates the adhesive’s effectiveness in bonding to teeth, generally dentin, which is essential to ensure dental restorations’ clinical performance and longevity [76]. Additionally, some studies assessed other physicochemical properties like flexural strength, modulus of elasticity, nanoleakage, viscosity, water sorption, and solubility.

The studies reviewed showed that, when natural antibacterial compounds were used at appropriate concentrations, they did not significantly impair essential adhesive properties. Only one study was a complete exception, as incorporating tt-farnesol [18] significantly reduced the degree of conversion and bond strength in all concentrations tested. As for nisin, the two studies included in this review reported slightly different outcomes in terms of what concentrations were able to preserve the adhesive’s bond strength since Su et al. (2018) reported that the incorporation of 3% (*w*/*v*) nisin decreased the microtensile bond strength, while Zhao et al. (2020) found that the same concentration did not affect it.

Additionally, EGCG was found to increase bond strength at 100 μg/mL and even more at 200 μg/mL [35]. Several studies have reported the ability of EGCG to inhibit matrix metalloproteinases (MMPs) [77,78]. MMPs are proteolytic enzymes that break down exposed collagen fibers within the dentin, resulting in the degradation of the hybrid layer and, consequently, decreased bond strength and durability [79]. Therefore, incorporating EGCG into dental adhesives could be a promising method for enhancing restoration longevity, as it combines antibacterial properties with a reduced degradation of the adhesive interface.

### 4.3. Concentration Dependency

The effectiveness of the antibacterial action and the maintenance of adhesive properties were closely linked to the concentration of the natural compounds used. This concentration dependency was evident across all studies that tested multiple concentrations.

Du et al. (2012) and Su et al. (2018) found that incorporating natural substances only exhibited significant antibacterial effectiveness above specific concentrations, in alignment with the rest of the studies reviewed, which indicated that higher quantities typically result in increased antimicrobial activity. However, these elevated concentrations were reported to negatively impact the adhesive’s integrity and bonding performance, whereas lower doses tended to preserve these properties [39,41,42,44,46].

Therefore, this indicates a delicate balance between achieving sufficient antimicrobial efficacy and retaining the essential physicochemical properties of the adhesive. It also accentuates the importance of evaluating both the mechanical and biological characteristics of the dental adhesive and multiple doses of the natural substances to identify an optimal concentration that successfully meets both criteria [80].

### 4.4. Long-Term Performance

Although the short-term antibacterial effects and adhesive properties were generally favorable, the long-term performance of these modified adhesives remains a significant challenge. A dental adhesive with immediate antibacterial activity may be able to reduce the residual bacteria present in the cavity; however, secondary caries often develop due to bacterial infiltration over time and the prolonged cumulative effects of their metabolism at compromised bond interfaces [81]. Aligned with that, it is also crucial that the adhesive can retain its mechanical properties over time, ensuring the stability and long-term clinical performance of restorations [82].

In this review, the long-lasting antibacterial properties of the modified adhesives were tested in some studies after 7 days [44], 1 month [35], 4 months [36], and 10,000 cycles (simulating 1 year of clinical physiological aging) [39]. Overall, the antimicrobial activity decreased but remained effective compared to the control, particularly at higher concentrations, except for EGCG [35], where effectiveness did not significantly decline after aging. Tt-farnesol’s performance was also evaluated after 5 days, but its incorporation severely compromised the adhesive’s properties, rendering it irrelevant [18]. As for the long-term assessment of the physicochemical properties of the adhesives, 500 μg/mL of quercetin [39], EGCG [35], and CMC [45] were successful in retaining bond strength after 1 month, 6 months, and 10,000 cycles of aging, respectively; quercetin also retained nanoleakage levels after that period. In contrast, Peralta et al. (2013) documented a reduction in bond strength over 6 months for the adhesive containing *Butia capitata* essential oil. In contrast, Ribeiro et al. (2021) noted a similar decrease for the adhesive containing apigenin + tt-farnesol after 1 year. Additionally, apigenin and the combination of apigenin + tt-farnesol increased nanoleakage over time [38].

These findings underscore the need for further investigation of the long-term performance of these natural compounds when incorporated into dental adhesives since a prolonged antibacterial activity is crucial for reducing the occurrence of secondary caries. It is also relevant to further evaluate how these natural compounds interact with the adhesive matrix over extended periods and under clinical conditions, as the decrease in bond strength over time may limit the practical application of these compounds unless formulations can be adjusted to mitigate such effects.

### 4.5. Biocompatibility and Safety

Although not explicitly detailed in the summary, a key issue in incorporating natural compounds into dental adhesives is ensuring their biocompatibility and safety for patients. Natural does not always equate to safe, and introducing new compounds into dental materials requires thorough testing to confirm they do not cause adverse reactions or compromise patient health in other ways, especially for agents with bactericidal activity [83]. Despite dentin acting as a barrier, its permeability increases when in close proximity to the dental pulp, making it crucial to assess potential toxic effects when developing novel dental materials, particularly regarding dental pulp cells, as they more accurately represent the in vivo target cells [84,85].

Only two studies in this review evaluated the cytotoxicity of the modified dental adhesives. Yang et al. (2017) conducted an MTT (3-(4, 5-dimethylthiazolyl-2)-2, 5-diphenyltetrazolium bromide) assay using human gingival fibroblast cells, finding no significant difference in cell viability after quercetin was incorporated at 100 and 500 μg/mL concentrations. Similarly, Rezaeian et al. (2019) found that 5 wt% thymol maintained cell viability; however, this assay was performed using mouse fibroblast cells. These results indicate these natural compound’s low cytotoxicity and acceptable biocompatibility for clinical use. Although not assessed in the studies reviewed, most other compounds have undergone cytotoxicity testing in other investigations [61,86,87,88,89].

This systematic review suggests that natural antibacterial agents could offer a more biocompatible and less cytotoxic alternative to synthetic chemicals [90,91,92]. However, further research is necessary to fully confirm the cytocompatibility and safety of incorporating these compounds into dental adhesives.

### 4.6. Clinical Implications

The findings from these studies carry significant clinical relevance. Incorporating antibacterial properties into dental adhesive systems holds promise for reducing bacterial colonization around restorations, potentially lowering the incidence of secondary caries and extending the longevity of restorations [8,16]. However, successfully applying these findings in practice requires balancing antibacterial effectiveness and the adhesive’s integrity and performance, ensuring long-term durability [73]. Additionally, ensuring a sustained antimicrobial effect and cytocompatibility is crucial [81,83]. Meeting these criteria is essential for the successful development of formulations before they can be adopted in clinical settings [80].

This review also contributes to the research on developing more natural, sustainable, and biological dental materials aligned with a holistic approach to dentistry [93,94]. Integrating these natural antibacterial substances into existing formulations could be a promising strategy for enhancing the therapeutic properties of adhesives.

### 4.7. Limitations

Overall, the present systematic review underscored the promising role of natural compounds in providing antibacterial effectiveness when incorporated into dental adhesive systems. Hence, the tested hypothesis was accepted, and the research question addressed, as the reviewed studies consistently demonstrated that incorporating natural antibacterial agents such as EGCG, chitosan, quercetin, thymol, and nisin into either experimental or commercial dental adhesive systems effectively provided antimicrobial activity without adversely affecting the adhesive’s physicochemical properties.

However, this evidence should be interpreted cautiously due to the high risk of bias across all studies. Despite this limitation, these articles were included in the review because the specific bias domains they failed to address—D3: Sample size calculation and D7: Operator blinded—were judged as unlikely to significantly impact the results or compromise the validity of the findings, especially in comparison to the other analyzed bias domains.

Furthermore, many studies still need to assess the long-term performance and cytotoxicity of the modified adhesives, as well as their antibacterial activity against a broader spectrum of microorganisms and in tooth substrates. Therefore, despite the promising findings, it is clear that further research is required to address these limitations.

## 5. Conclusions

The utilization of natural antibacterial compounds has been evidenced to be effective in inhibiting bacterial growth and biofilm formation, particularly in the case of Streptococcus mutans. Integrating these compounds into dental adhesives represents a promising approach for reducing the prevalence of secondary caries, thereby enhancing the restorations’ durability and longevity while maintaining the adhesive bond’s integrity. The present systematic review indicates that 200 μg/mL of EGCG, 500 μg/mL of quercetin, and 20 mg/mL of CMC have the potential to balance effective antimicrobial activity while maintaining the adhesive’s integrity and stability. Nevertheless, further research is necessary to address current limitations, to translate in vivo current in vitro results, and to ascertain the long-term performance, cytotoxicity, and antibacterial activity of these modified adhesives against multiple microorganisms.

## Figures and Tables

**Figure 1 polymers-16-03217-f001:**
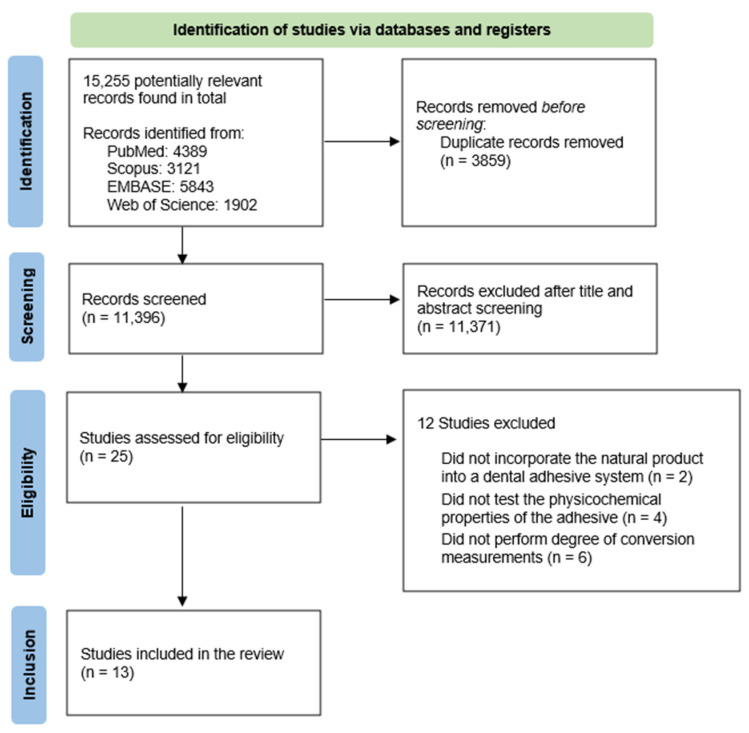
Flowchart outlining the study selection process (as described in the PRISMA 2020 statement guidelines).

**Figure 2 polymers-16-03217-f002:**
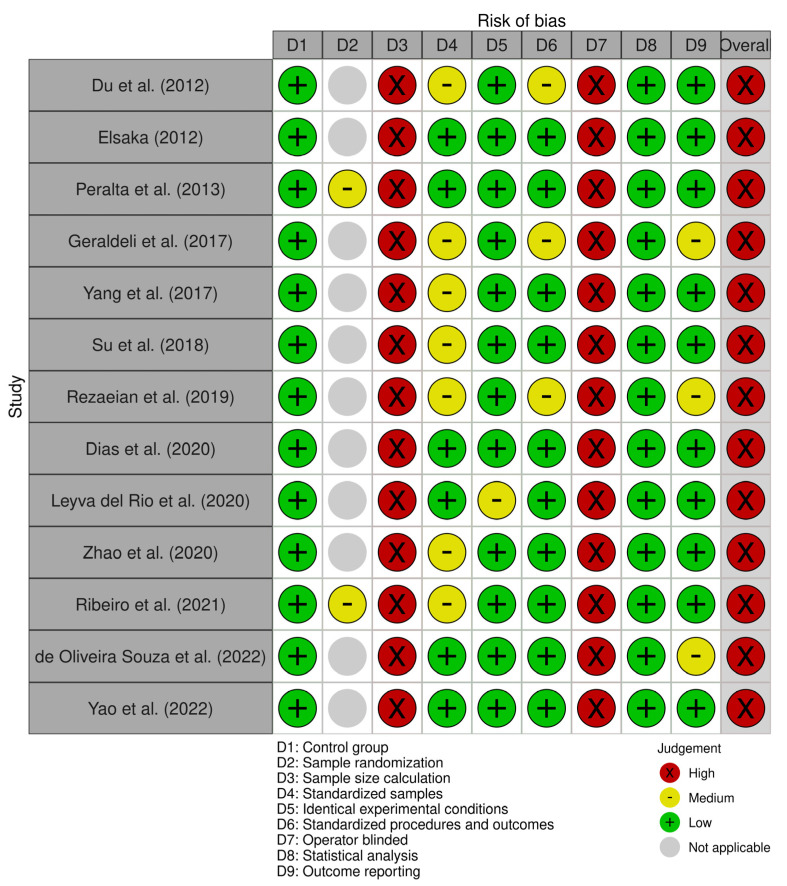
Traffic light plot of the studies’ risk of bias (RoB) analysis [18,35,36,37,38,39,40,41,42,43,44,45,46].

**Figure 3 polymers-16-03217-f003:**
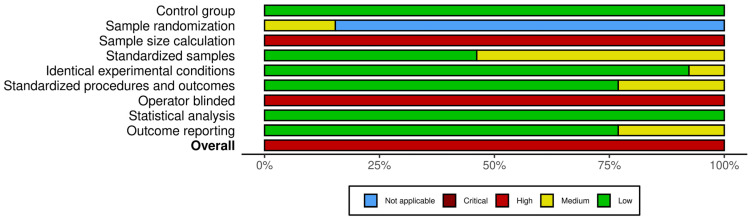
Summary plot of the studies’ risk of bias (RoB) analysis.

**Table 1 polymers-16-03217-t001:** Search algorithm used in each database.

Database	Search Algorithm	Search Date
PubMed	(antibacterial OR antimicrobial OR antibiotic OR anticaries OR antiseptic OR anticaries) AND (“Dental Adhesive” OR bond OR primer OR adhesive) AND (dental OR “restorative dentistry” OR “dental restoration” OR “adhesive dentistry”) NOT (review)	17 July 2024
Scopus	ALL ((antibacterial OR antimicrobial OR antibiotic OR anticaries OR antiseptic OR anticaries) AND (“Dental Adhesive” OR bond OR primer OR adhesive) AND (dental OR “restorative dentistry” OR “dental restoration” OR “adhesive dentistry”) AND NOT (review))	17 July 2024
EMBASE	(antibacterial OR antimicrobial OR antibiotic OR anticaries OR antiseptic OR anticaries) AND (“Dental Adhesive” OR bond OR primer OR adhesive) AND (dental OR “restorative dentistry” OR “dental restoration” OR “adhesive dentistry”) NOT review	18 July 2024
Web of Science	ALL = ((antibacterial OR antimicrobial OR antibiotic OR anticaries OR antiseptic OR anticaries) AND (“Dental Adhesive” OR bond OR primer OR adhesive) AND (dental OR “restorative dentistry” OR “dental restoration” OR “adhesive dentistry”) NOT (review))	18 July 2024

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
