# Peer review of "Natural Antibacterial Compounds with Potential for Incorporation into Dental Adhesives: A Systematic Review"

_polymers, 2024, doi:10.3390/polym16223217_

Round 1
Reviewer 1 Report
Comments and Suggestions for Authors
Dear Authors,
Your manuscript is well-written and provides a comprehensive review of the topic. However, I suggest that you justify the inclusion/selection of the papers with such a high risk of bias. It is important to address how this may impact negatively the validity of your findings and explain the rationale for their inclusion despite this limitation.
Author Response
Comments 1: Your manuscript is well-written and provides a comprehensive review of the topic. However, I suggest that you justify the inclusion/selection of the papers with such a high risk of bias. It is important to address how this may impact negatively the validity of your findings and explain the rationale for their inclusion despite this limitation.
Response 1: We are grateful for your observation. We have endeavoured to address the concerns you have raised. Accordingly, a new subsection, entitled "4.7", has been introduced in the Discussion section. In the section entitled "Limitations" (page 16, paragraph 4), this article's findings are presented cautiously, given the risk of bias in two specific domains across all studies. Notwithstanding this limitation, the articles in question were included in the review on the grounds that the specific bias domains they failed to address – D3: Sample size calculation and D7: Operator blinded – were deemed unlikely to exert a significant influence on the results or compromise the validity of the findings, particularly in comparison to the other analysed bias domains, given the quantitative nature of the outcome of the analysed research. Therefore, in this context, including articles with a high risk of bias is justified as they can contribute to a better understanding of the subject matter.
Reviewer 2 Report
Comments and Suggestions for Authors
1. It is recommended to proofread the entire text carefully to check for spelling, grammatical and punctuation errors, and to ensure that the text is concise and smooth. It is recommended that you find an English editor to polish it before submitting.
2. The abstract should clearly indicate the research methods of this systematic review, such as the screening criteria for included studies, database search strategies, etc.
3. In the introduction, the historical development of dental adhesives is described too briefly. The authors should detail the evolution of dental adhesives, especially trends related to antimicrobials. At the same time, the authors should more clearly describe the advantages of using natural antimicrobials, such as biocompatibility, sustainability, and cost-effectiveness compared with synthetic antimicrobials, and include a description of the advantages of enhanced natural antimicrobials.
4. In the definition part of the research problem, the description of the population (P) in the PICO framework is not precise enough, and the author should reflect the research object more accurately.
5. In the search strategy section, all keywords and Boolean operators used should be listed, and the time range of the database search should be stated so that the content of Table 1 can be presented more completely, such as adding all keywords and Boolean operators used. and the time range for database searches.
6. Inclusion and exclusion criteria should be clearer, e.g., detailing which types of studies will be excluded. It is recommended to add more details about the exclusion criteria.
The discussion section should provide a more in-depth analysis of the antibacterial mechanisms of different natural antibacterial agents and a comparison of their advantages and disadvantages as well as a more in-depth analysis of the advantages and disadvantages. For example, some antibacterial agents may have broad-spectrum antibacterial effects, while others may be effective against specific bacteria. More effective. In addition, the potential risks, and challenges of adding natural antibacterial agents to dental adhesives should also be discussed and analyzed. For example, some natural antibacterial agents may release harmful substances after long-term use or affect the mechanical properties of the adhesive. performance.
Comments on the Quality of English Language
1. It is recommended to proofread the entire text carefully to check for spelling, grammatical and punctuation errors, and to ensure that the text is concise and smooth. It is recommended that you find an English editor to polish it before submitting.
2. The abstract should clearly indicate the research methods of this systematic review, such as the screening criteria for included studies, database search strategies, etc.
3. In the introduction, the historical development of dental adhesives is described too briefly. The authors should detail the evolution of dental adhesives, especially trends related to antimicrobials. At the same time, the authors should more clearly describe the advantages of using natural antimicrobials, such as biocompatibility, sustainability, and cost-effectiveness compared with synthetic antimicrobials, and include a description of the advantages of enhanced natural antimicrobials.
4. In the definition part of the research problem, the description of the population (P) in the PICO framework is not precise enough, and the author should reflect the research object more accurately.
5. In the search strategy section, all keywords and Boolean operators used should be listed, and the time range of the database search should be stated so that the content of Table 1 can be presented more completely, such as adding all keywords and Boolean operators used. and the time range for database searches.
6. Inclusion and exclusion criteria should be clearer, e.g., detailing which types of studies will be excluded. It is recommended to add more details about the exclusion criteria.
7. The discussion section should provide a more in-depth analysis of the antibacterial mechanisms of different natural antibacterial agents and a comparison of their advantages and disadvantages as well as a more in-depth analysis of the advantages and disadvantages. For example, some antibacterial agents may have broad-spectrum antibacterial effects, while others may be effective against specific bacteria. More effective. In addition, the potential risks, and challenges of adding natural antibacterial agents to dental adhesives should also be discussed and analyzed. For example, some natural antibacterial agents may release harmful substances after long-term use or affect the mechanical properties of the adhesive. performance.
Round 2
Reviewer 2 Report
Comments and Suggestions for Authors
The authors already revised their paper point-by-point according to reviewers’ comments and tried their best to improve their manuscript. In my opinion, this article can be published in this journal.